# Analysis of football research trends using text network analysis

**Jongwon Kim** *

London Sport Institute, School of Science and Technology, Middlesex University, London, United Kingdom

* jongwonkim1229@gmail.com

**Data Availability Statement:** All relevant data are within the manuscript and its Supporting Information files.

**Funding:** The author(s) received no specific funding for this work.

## Abstract

This study was aimed to identify football research trends in various periods. A total of 30,946 football papers were collected from a representative academic database and search engine, the 'Web of Science'. Keyword refinement included filtering nouns, establishing synonyms and thesaurus, and excluding conjunctions, and the Cyram's Netminer 4.0 software was used for network analysis. A centrality analysis was conducted by extracting the words corresponding to the top 2% of the main research topics to obtain the degree and eigenvector centralities. The most frequently mentioned research keywords were injury, performance, and club. Keyword performance showed the highest degree centrality (0.294) and keyword world and cup showed the highest eigenvector centrality (0.710). The keyword with the highest eigenvector degree changed from injury in the 1990s and world in the 2000s to cup since the 2010s. Although various studies on football injuries have been conducted, research on the sport itself has recently been conducted. This study provides fundamental information on football trends from research published over the past 30 years.

## Introduction

The match between Argentina and France in the FIFA World Cup Qatar 2022 final was watched by 1.5 billion people around the world, and the cumulative number of World Cup viewers exceeded 7 billion, which is a huge figure considering that the world's population is approximately 8 billion [1]. Much effort has been devoted by football players and coaches as well as researchers in academia to improve the player's performance and support the entire football industry. For example, numerous studies have been conducted on indicators of player's technique [2, 3], tactic [4, 5], physical ability [6, 7], physiological function [8, 9], psychological state [10, 11], among others to better understand football.

A survey of existing studies should be conducted objectively to identify current research through systematic analysis of the corresponding findings [12]. Research trend analysis contributes to academic development by critically and objectively accessing existing studies and providing a basis for topics and directions to promote further advances. Several researchers have comprehensively analyzed football studies using systematic reviews and meta-analyses [9, 13, 14]. However, such approaches impose a strict methodology [15], failing to adequately

**Competing interests:** The authors have declared that no competing interests exist.

cover an entire research area [16]. Consequently, accommodating an entire body of research is increasingly difficult when adopting available literature review methods [12].

Social network analysis has recently been widely used for unveiling research trends in specific topics [17]. This type of analysis involves a measurement technique for evaluating data through their structures, interaction, and relationships [18]. In particular, text network analysis is an excellent artificial intelligence technique to describe and analyze relationships between words in text through a network [19]. To generate a specific network, the scope of the dataset should be first decided [20]. Data collecting techniques depend on data gathering principles such as collection from specific journals [21], conferences [22], institutions, universities, or nations [23], or a particular scientific domain [20]. Published research papers are generally chosen based on such principles [20]. The corresponding techniques have been used for analyzing co-authorship in various fields, such as medical science [24], education [25], mining [20], and health [26]. Regarding football research using social network analysis, Kirkendall and Krustrup [27] explored papers on female and professional football players and identified that studies about female players accounted for approximately 20% of all football research and approximately 15% of studies on professional player.

Unlike conventional literature analysis that focuses on qualitative analysis of available data, text network analysis allows to quantitatively explore the knowledge structure of a research domain [12]. Network-based reviews have various advantages as a complement to traditional analytical methods. Remarkably, network text analysis can overcome limitations of reading and manually classifying numerous papers. In this study, we aimed to determine the main keywords in football research using all the available research topics based on network theory and understand the relationships between those keywords. Furthermore, we intended to identify the trends of major keywords over time and derive future research directions for football.

## Materials and methods

### Study overview

First, we collected papers for this study and then extracted their text data. The extracted text data were refined and preprocessed to suppress noise and irrelevant information. A network was constructed by defining the relationships between text elements such as words, sentences, and documents. Then, network analysis techniques were applied to explore the structure and characteristics of the text network. Finally, a visualization technique was applied to facilitate the understanding and interpretation and obtain insights into patterns within the text. The procedure followed in this study is illustrated in Fig 1.

### Data collection

We collected data from the Web of Science, a representative academic database and search engine, to analyze research topics related to football (soccer) from published work. As the term for football varies in some countries, we included both two football and soccer ("football" OR "soccer" on the search engine) for paper search. Papers containing "football" or "soccer" in the title or abstract were included in our study. In addition, only papers written in English were considered. Overall, we obtained 46,302 papers, from which 8,486 duplicates were excluded. Through search of 69 keywords, such as American, Gaelic, rugby and NFL, in the paper title and abstract, 6,269 papers were excluded because they were not related to football (soccer) despite using this term. For example, studies on American football, Gaelic football, and Canadian football were excluded despite using the term football. Moreover, 601 papers published before 1990 were excluded because most of them did not contain an abstract, impeding to identify whether the main topic was football (soccer). Finally, 30,946 papers were considered

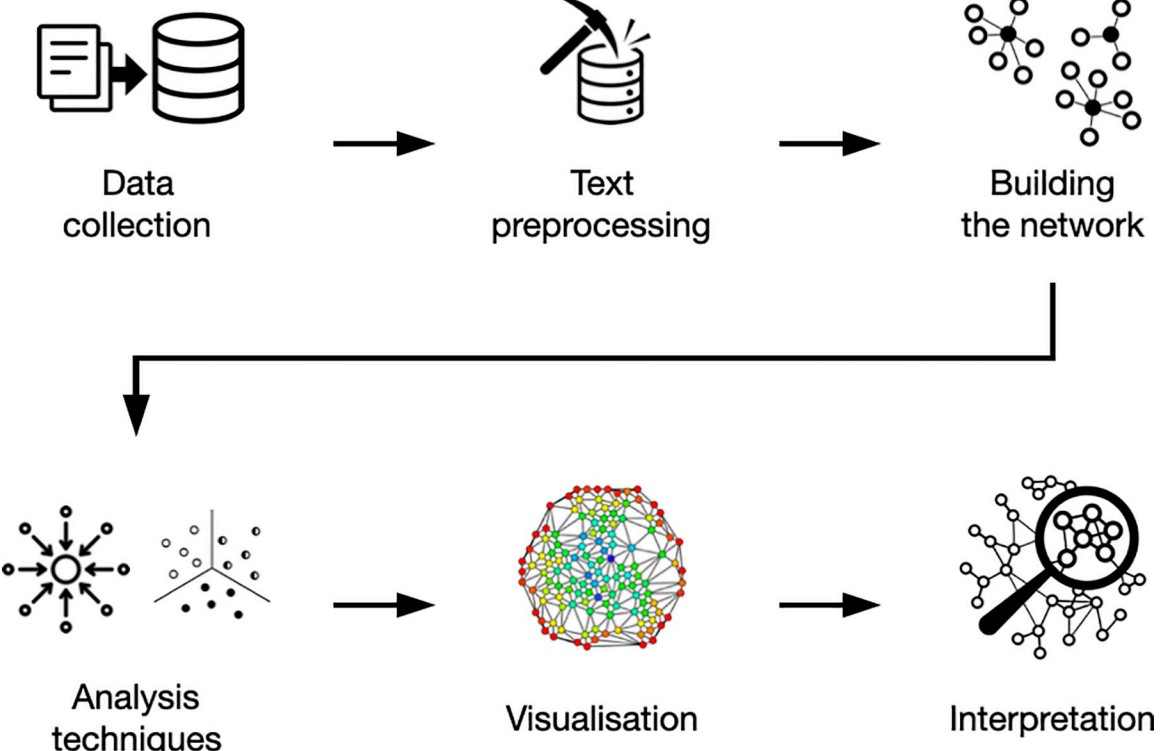

**Fig 1. Procedure for text network analysis adopted in this study.**

for this study. The last date of paper search was February 5, 2023. We provide a S1 Raw data (in csv format) including the authors, publication year, title and abstract of the collected papers.

## Data preprocessing

Refining collected data is important for accurate network analysis [28]. First, we extracted nouns from each paper, and nouns related to the same word, similar words, and exception words were refined. If an adjective was nominalized or similar words could be expressed as one word, they were designated as a synonym. For example, "character", "characteristic", and "characterization" can be deemed similar. Therefore, all these words were pre-registered as "character" in a thesaurus (i.e., a dictionary of similar words). Similarly, "player" and "athlete" were merged into "player" and registered in a synonym dictionary. Moreover, if various words could be expressed as one term, they were defined in advance in the dictionary. For example, to solve the problem of "English", "premier", "league" appearing separately and recognized as three words, "English Premier League" was predefined as one term, and its abbreviation EPL was defined as a synonym. Similarly, the word dictionary was populated with compound words such as "football player", "performance analysis", and "world cup". On the other hand, words with weak meaning as research topics were excluded. Such words included day, end, fact, help, and item were excluded from the analysis because they may fail to clearly represent a research topic when considered as independent words. In addition, preprocessing for network analysis considered abbreviations and spacing using the Cyram Netminer 4.0 Filter and Dictionary functions.

## Network construction

After extracting representative text from research papers, the research keywords corresponding to each research paper was arranged into two-mode matrix data. The two-mode data constructed in this way were used to calculate the cooccurrence of the main topics and express those topics in one mode for network analysis between keywords [29]. Among the refined words and terms, "soccer player" appeared in 4,052 papers, "football player" appeared in 2,625 papers, and the Gini coefficient, a measure of statistical dispersion that represents the inequality of a distribution (score range of 0~1 with 1 indicating perfect distribution) [30], was 0.9 without considering the network. To analyze the network between words, two-mode data were expressed in one mode (word × word). We also obtained the Ochiai and Salton coefficients that quantify the similarity between sets or vectors [30], and self-loops generated during data conversion were removed. The Ochiai coefficient has a value between 0 and 1, and it helps clustering highly related and similar words, thereby creating a network that can be simplified by considering high-weighted relationships [18].

## Statistical analysis

**Centrality analysis.** A centrality analysis was conducted by extracting only words corresponding to the top 2% of the main research topics (262 out of 13,097 nodes). Among the various existing centrality analyses, we calculated the degree and eigenvector centralities [29]. The degree centrality allows to measure the number of nodes directly connected to the main research topics and grasp direct influences. The degree centrality reaches its maximum when connected to all nodes except itself (Formula 1). The degree centrality ranges from 0 to 1, and a larger value indicates a higher influence [29]. The eigenvector centrality (Formula 2) extends the degree centrality. It not only reflects the node connections, but its value increases when connected nodes have a high centrality.

$$Degree\ Centrality = \frac{d(n_1)}{g - 1}$$

$$g = total\ number\ of\ nodes,\ d(n_1) = degree\ of\ node\ n$$

Formula 1. Degree centrality

Eigenvector Centrality of node vi: $c_e(v_1) = \frac{1}{\gamma} \sum_{j=1}^{n} A_{j,\ iC_e}(vi)$

$$\gamma = eigen\ value,\ A = 1 - adjacency\ matrix,\ n = total\ number\ of\ nodes$$

Formula 2. Eigenvector centrality

**Clustering analysis.** We also conducted a clustering analysis to understand the entire network and subgroups of main research topics. The number of clusters for a substructure and connections between clusters were analyzed. We determined a subgroup by calculating the modularity coefficient (0–1) based on greedy heuristics of hard clustering [31, 32]. A coefficient closer to 0 indicates more difficulty to distinguish substructures, while a coefficient closer to 1, indicates clearer substructures that can be easily distinguished. In other words, a higher modularity indicates a higher connection density in a substructure and lower connection density between substructures [33]. We used Cyram's Netminer 4.0 for network analysis and IBM SPSS 26 and Microsoft Office Excel 2016 for intermediate statistical analyses.

## Results

### Research characteristics

The number of published papers on football research is rapidly increasing over time (Fig 2). Only 1,405 papers were published during the 1990s, while 4,713 and 15,315 papers were published in the 2000s and 2010s, respectively. In the 2020s, 9,513 papers have already been published until 2022.

Table 1 lists the most frequently mentioned research keywords on football research papers between 1990 and 2022. Excluding keywords not indicating a specific research topic, such as "football", "soccer", "player", and "research", the most frequent keyword was "injury", followed by "performance", "club" and "training".

Both the amount and topics of studies have changed over time. Regarding the number of papers, only 6.4% (n = 2,006) of the papers were published in the 1990s, while the number of published papers increased exponentially to 14.9% (n = 4,713) in the 2000s and 48.5% (n = 15,315) in the 2010s, and approximately 30% (n = 9,513) of the papers were published in the first 3 years of the 2020s. The main research topics were "injuries" and "university" before 2000, "performance" and "training" since 2000, and various topics such as "youth", "women", and "clubs" have recently emerged.

### Degree and eigenvector centralities

Keyword "performance" (0.294) reached the highest degree centrality from the collected football research papers, followed by "injury" (0.118) and "training" (0.118). Meanwhile, the highest eigenvector centrality was obtained by keywords "world" (0.710) and "cup" (0.710), followed by "analysis" (0.068) and "club" (0.008).

A cohesive structure analysis (modularity of 0.662) revealed seven clusters (groups G1-G7) regardless of the keyword ranking (Fig 3). G1 included keyword "performance" with the highest degree centrality and involved research on topics such as performance level, change, and relationship. G2 was related to analyses of world cups or football clubs, while G3 mainly considered injuries, especially that of the knee ligament. In addition, G4 was related to training and G5 included research keywords of populations such as elite, youth, male and female. Finally, G6 included experiments conducted in schools or universities, especially regarding concussions, and G7 was related to exercise and game.

### Change in research keywords over time

To identify changes in research keywords over time, we conducted a text network analysis in four periods: 1990s (1990–1999), 2000s (2000–2009), 2010s (2010–2019), 2020s (2020–2022).

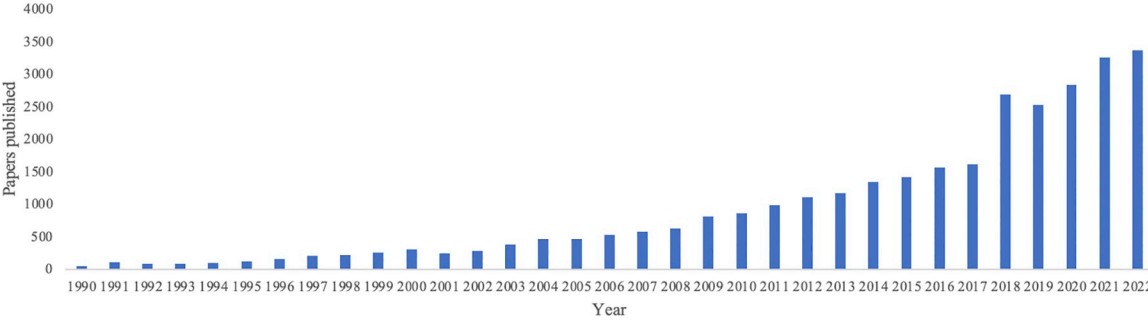

**Fig 2. Number of football research papers published by year.**

**Table 1. Frequency of keywords appearing in football research papers.**

| Rank | | Number of papers |
|------|----------|------------------|
| 1 | Injury | 3,107 |
| 2 | Performance | 2,751 |
| 3 | Club | 2,121 |
| 4 | Training | 1,934 |
| 5 | Elite | 1,713 |
| 6 | Analysis | 1,643 |
| 7 | Youth | 1,448 |
| 8 | Assessment | 1,253 |
| 9 | Game | 1,249 |
| 10 | Female | 1,083 |
| 11 | Case | 1,028 |
| 12 | University | 916 |
| 13 | Muscle | 866 |
| 14 | Concussion | 842 |
| 15 | Strength | 840 |
| 16 | Exercise | 804 |
| 17 | Match | 760 |
| 18 | Activity | 748 |
| 19 | Relationship | 745 |
| 20 | Knee | 724 |
| 21 | Risk | 722 |
| 22 | Comparison | 711 |
| 23 | Change | 698 |
| 24 | Male | 689 |
| 25 | Program | 686 |
| 26 | Association | 670 |
| 27 | School | 656 |
| 28 | World | 651 |
| 29 | Ligament | 604 |
| 30 | Development | 598 |

Periods 1990s, 2000s, 2010s, and 2020s included 1,405 (4.5%), 4,713 (15.2%), 15,315 (49.5%), and 9,513 (30.7%) papers, respectively. Keyword "injury" was the most frequent in the 1990s (14.1%), and its incidence gradually decreased to 11.5% in the 2000s and 9.8% in the 2010s to finally appear as the second most frequent keyword with 8.2% in the 2020s. On the other hand, keyword "performance" was the fourth most frequent (4.2%) in the 1990s and gradually increased to 5.6% in the 2000s and 9.3% in the 2010s to finally be the most frequent with 10.2% in the 2020s. Interestingly the keywords "university" and "knee" steadily decreased, while "female" dramatically increased.

Research published in the 1990s was divided into seven clusters (modularity of 0.672) comprising various keywords (Fig 4). The keywords included "training" (G1), "injury" (G2), "exercise" (G3), "university" (G4), "case" (G5), "game" (G6), and "performance" (G7). Regarding the degree centrality, the main connected keywords were "injury" (0.176), "performance" (0.118), "strength" (0.088), "adolescent" (0.088), "club" (0.088), and "robot" (0.088). Regarding eigenvector centrality, "injury" (0.651), "ligament" (0.500), and "high school" (0.303) showed the highest values.

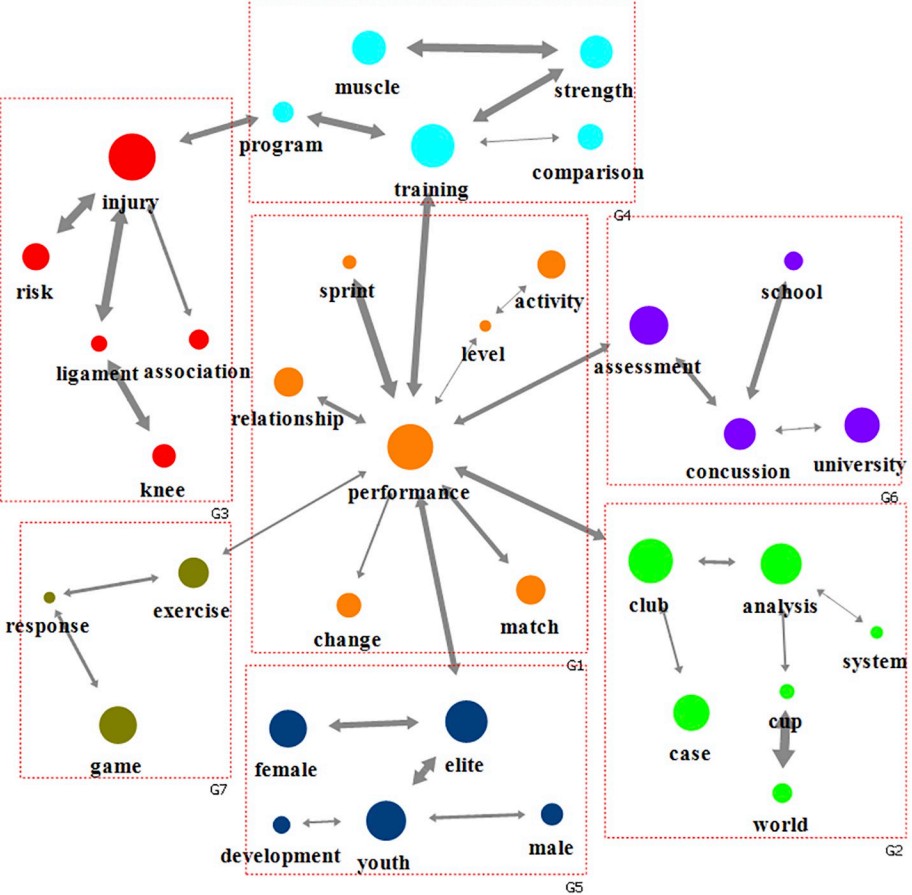

**Fig 3. Analysis of text network for football research keywords over all study periods.**

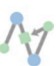

In the 2000s, the number of papers more than doubled compared with the 1990s, and six clusters were obtained (modularity of 0.662) (Fig 5). The keywords included "injury" (G1), "analysis" (G2), "performance", "university", and "assessment" (G3), "training" (G4), "school" (G5), and "case" (G6). Regarding the degree centrality, the main connected keywords were "injury" (0.171), "performance" (0.114), "strength" (0.114), and "assessment" (0.114). Regarding the eigenvector centrality, "world" (0.707), "cup" (0.707), and "case" (0.043) showed the highest values.

In the 2010s, the largest number of papers (15,315) was published, and six clusters (modularity of 0.678) were formed, as in the 2000s (Fig 6). The cluster topics included "training" (G1), "assessment of concussion at university" (G2), "injury" (G3), "club" and "analysis" (G4), population characteristics such as "elite", "youth", "male", and "female" (G5), and "performance" (G6). Regarding the degree centrality, the main connected keywords were "performance" (0.206), "training" (0.147), "concussion" (0.118), and "injury" (0.118). Regarding the eigenvector centrality, "cup" (0.707), "world" (0.703), and "analysis" (0.078) showed the highest values.

In the 2020s, six clusters (modularity of 0.670) were formed and related to "training" and "game" (G1), "injury" (G2), "club" and "analysis" (G3), "performance" (G4), "elite", "youth", "male", "female", and "age" (G5), and "assessment" (G6) (Fig 7). Regarding the degree centrality, the main connected keywords were "performance" (0.265), "club" (0.118), "load" (0.118),

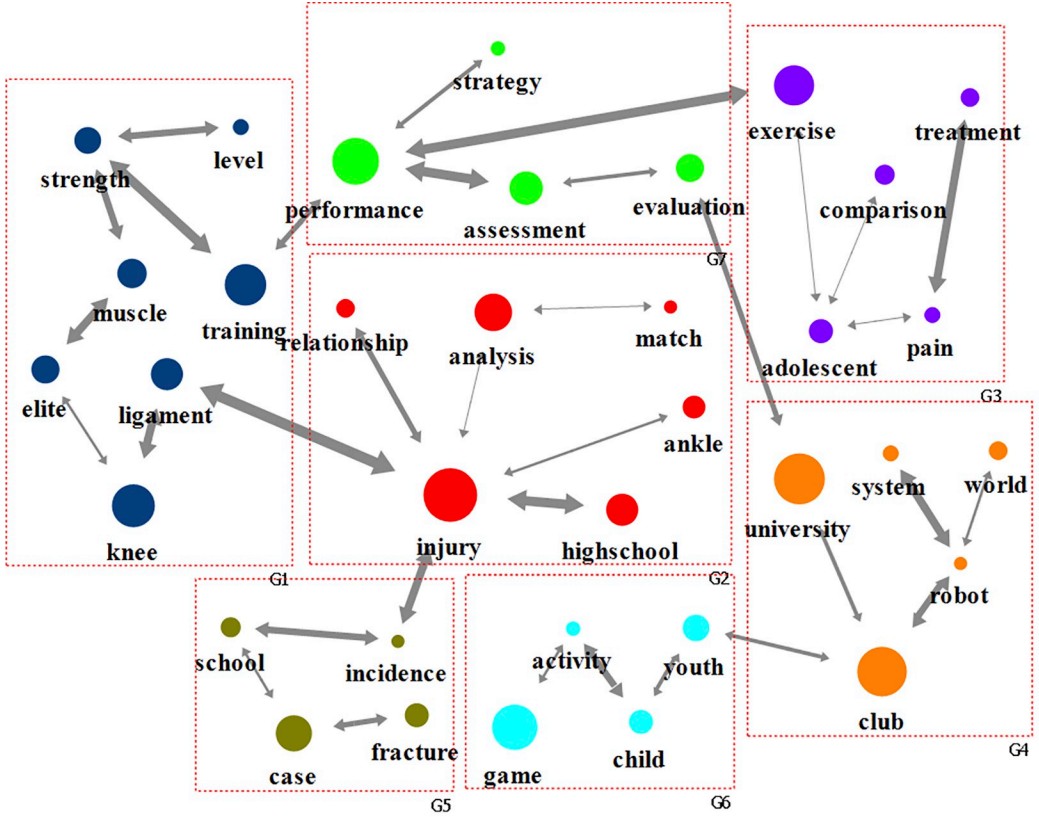

**Fig 4. Analysis of text network for football research keywords in 1990s.**

and "training" (0.118). Regarding the eigenvector centrality, the results were almost same as in the 2010s, with "cup" (0.707), "world" (0.073), and "analysis" (0.073) showing the highest values.

## Discussion

We applied network analysis to identify trends of the keywords in published football research papers over time. The most used research keyword over the past 30 years was "injury" (9.85%), followed by "performance" (8.72%), "club" (6.72%), "training" (6.13%), and "elite" (5.43%). Studies on injuries accounted for approximately 15% of all football research papers until the 2000s and gradually decreased over time. Our results, while similar to previous research [27] that identified injury as a frequently used research keyword in football, showed a contrasting outcome in the sense that the ratio of injuries over all the research keywords is decreasing over time. In the 2020s "injury" was ranked second (8.24%) after "performance" (10.22%). Most studies on injuries investigated the ankle in aspects such as pain [34, 35], recovery [36, 37], and stability [38, 39] of ankle fractures, sprain, or ligament tears in the 1990s. In addition, studies on the knee [40, 41] and muscle injury [42] have recently increased rapidly. These studies seem to have influenced the development of inspection tools. Before the development of testing tools (in the 1990s), research on body parts (e.g., ankles) whose injuries could be detected through surface anatomy was predominant. Since the 2000s, measurements in the knees and muscles along with magnetic resonance and ultrasonic imaging became available.

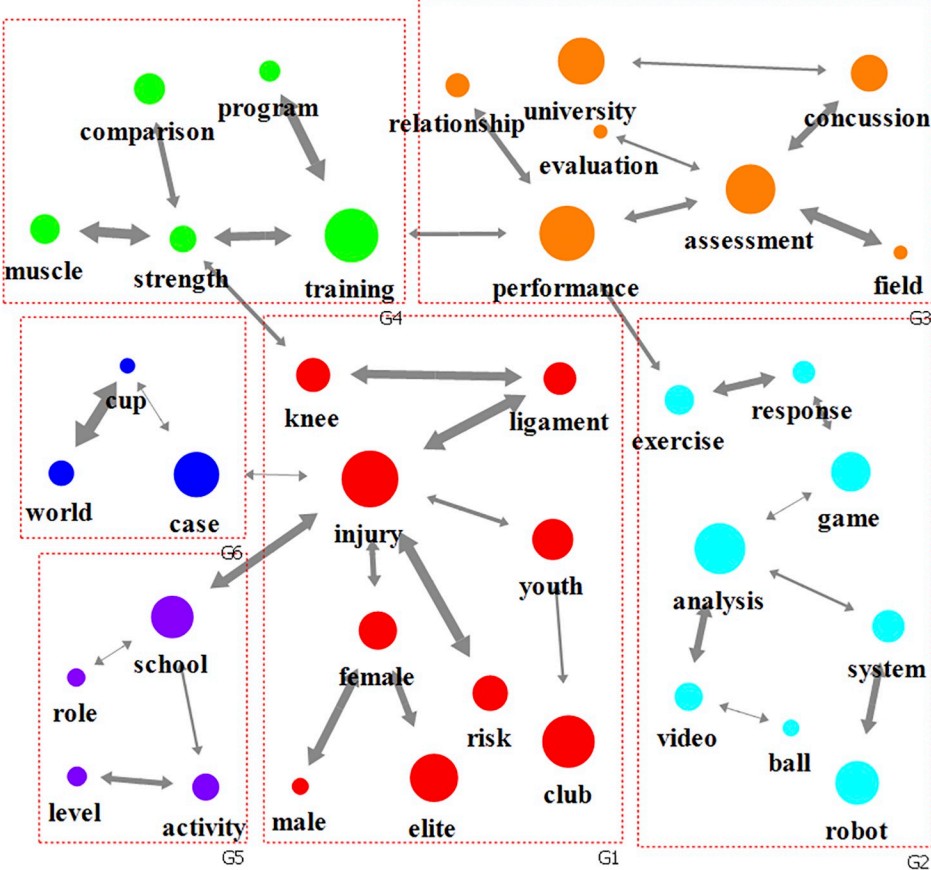

**Fig 5. Analysis of text network for football research keywords in 2000s.**

Unlike research on injuries, studies on football matches gradually increased over time. Among them, research on performance has increased rapidly, and after in the 2020s, it has been the most studied research topic, surpassing injury research. In the past, research on performance was conducted at universities or schools through laboratory measurements and experiments. Recently, research on adult competitiveness involving elite players is becoming more common. Research on performance mainly involves physical [7], technical [2], or tactical [5] performance. Unlike previous laboratory experiments, recent research has been conducted using field measurements mainly acquired from professional football leagues (e.g., EPL, La Liga, Bundesliga) and international competitions (e.g., World Cup, Europa League, Olympics). These research trends are likely driven by advancement of media and globalization, fostering professional sports and international competitions. As sports teams and national representatives increasingly require scientific research to increase their competitiveness, extensive research is being conducted on topics related to player's performance. In future research, studies focused on enhancing the performance of players and teams should be integrated into evolving research trends.

Along with changing research topics and samples, the amount of data used in research has also evolved. Unlike early studies that considered few research samples or case studies involving small groups, current research is conducted with numerous research samples, such as whole matches, and the number of research variables is increasing. In addition, in recent

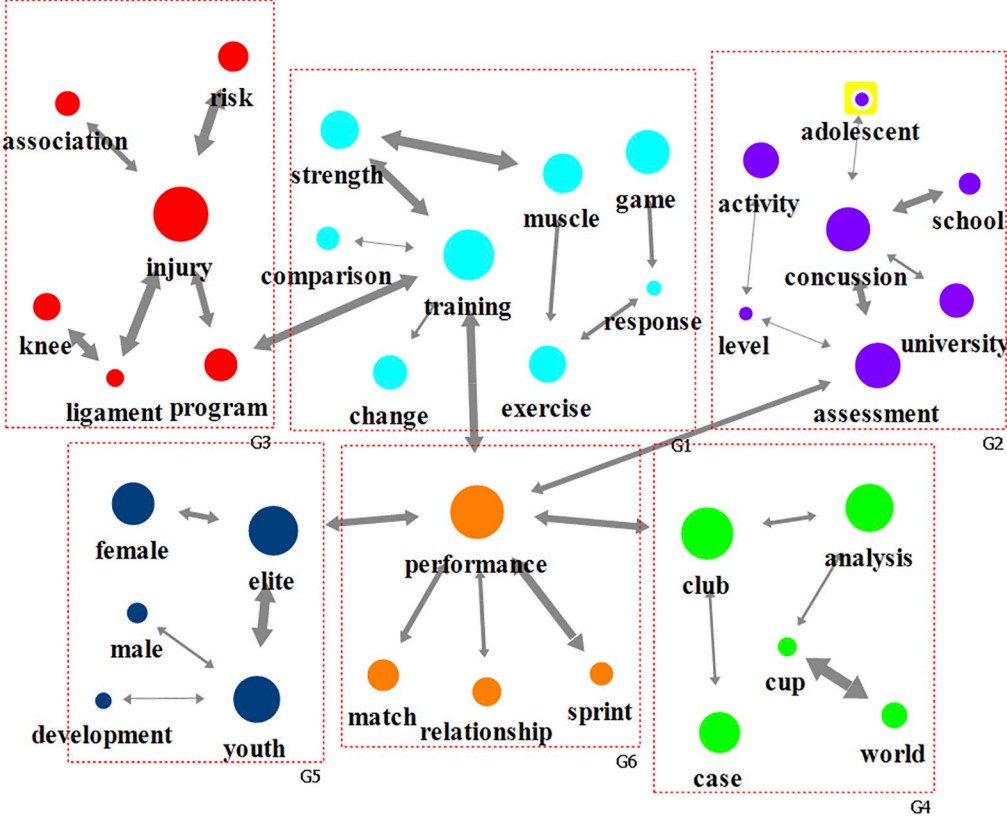

**Fig 6. Analysis of text network for football research keywords in 2010s.**

years, many studies using large datasets have been conducted by leveraging artificial intelligence [43, 44] and big data [45, 46]. Early studies were mainly conducted to obtain descriptive analysis results or compare results through simple experiments. Recently, research on causality and prediction is becoming prevalent.

As a result of investigating studies on football over the past 30 years, the research trends seem to have changed largely through three periods. These periods may have been characterized by 1) the possibility of experiments with the development of inspection tools, 2) the popularity of professional football leagues and international competitions, and 3) increase in the amount of collected research data with the advancement of science and technology. The word dictionary was difficult to generalize because researchers subjectively influenced the definition and similarity designation. Thus, some results may have been biased. For example, weak expressions to be research topics were excluded, such as "arrival", "axis", and "anything", and similar words were designated as synonyms, such as "character", "characteristic", and "characterization", indicating subjective judgement. In addition, the football keywords were difficult to categorize. For example, keyword "football performance" is difficult to categorize into "physiology", "football psychology", or "football medicine" to extract additional research context. Hence, we believe that the research flow of each football research area should be detailed through subdivisions such as "physiology", "football medicine", "football analysis", and "football marketing" rather than using only the generic world "football". Despite its limitations, this study allowed us to identify keywords and research trends in football research from over 30,000 papers spanning 30 years of research.

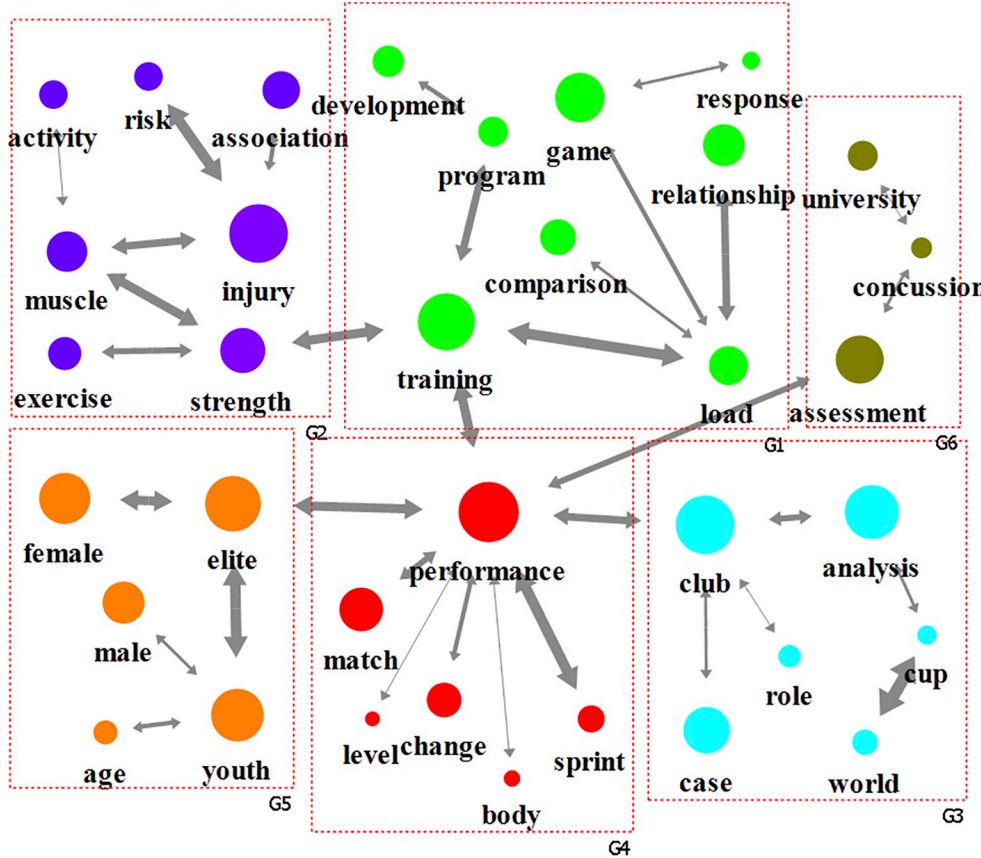

**Fig 7. Analysis of text network for football research keywords in 2020s.**

## Conclusion

We can draw the following conclusions from this study. First, the number of football research papers is increasing rapidly. Second, the most frequent research topic was "injury", but its incidence gradually decreased, with the most frequent keyword in the 2020s being "performance". Third, with the development of inspection tools, the popularity of professional football leagues, and the development of science and technology, the research topics have evolved. Although we faced limitations to determine exact research trends from the collected research papers, we obtained valuable insights into overall trends of football research published over the past 30 years and possible future trends.

## Supporting information

**S1 Raw data.**
(XLSX)

## Author Contributions

**Conceptualization:** Jongwon Kim.

**Formal analysis:** Jongwon Kim.

**Investigation:** Jongwon Kim.

**Methodology:** Jongwon Kim.

**Project administration:** Jongwon Kim.

**Resources:** Jongwon Kim.

**Supervision:** Jongwon Kim.

**Visualization:** Jongwon Kim.

**Writing – original draft:** Jongwon Kim.

**Writing – review & editing:** Jongwon Kim.

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
