## [Decision Letter · Decision Letter 0]

12 Jan 2024

PONE-D-23-39390Analysis of football research trends using text network analysisPLOS ONE

Dear Dr. KIM,

Thank you for submitting your manuscript to PLOS ONE. After careful consideration, we feel that it has merit but does not fully meet PLOS ONE’s publication criteria as it currently stands. Therefore, we invite you to submit a revised version of the manuscript that addresses the points raised during the review process.

We look forward to receiving your revised manuscript.

Kind regards,

Julio Alejandro Henriques Castro da Costa

Academic Editor

PLOS ONE

Reviewers' comments:

Reviewer's Responses to Questions

**Comments to the Author**

1. Is the manuscript technically sound, and do the data support the conclusions?

Reviewer #1: Yes

Reviewer #2: Partly

Reviewer #3: Yes

2. Has the statistical analysis been performed appropriately and rigorously? 

Reviewer #1: Yes

Reviewer #2: I Don't Know

Reviewer #3: Yes

3. Have the authors made all data underlying the findings in their manuscript fully available?

Reviewer #1: Yes

Reviewer #2: Yes

Reviewer #3: Yes

4. Is the manuscript presented in an intelligible fashion and written in standard English?

Reviewer #1: Yes

Reviewer #2: No

Reviewer #3: Yes

5. Review Comments to the Author

Reviewer #1: **Subject:** Manuscript Review for PONE-D-23-39390: "Analysis of Football Research Trends Using Text Network Analysis"

**Reviewer:**

#### Overview

The manuscript provides a comprehensive analysis of football research trends over the last thirty years using a text network analysis. The approach is innovative and the topic is relevant to the field.

#### Major Comments for Revision

1. **Methodology Clarity**: The methodology section, while thorough, could benefit from clearer explanations of the text network analysis process. A step-by-step breakdown would enhance the reader's understanding.

2. **Data Representation**: The manuscript presents a vast amount of data. However, the current visual representations (graphs, tables) could be improved for better clarity and impact.

3. **Comparative Analysis**: The study primarily focuses on keyword trends. Including a comparative analysis with other sports or academic disciplines could provide a richer context.

4. **Discussion of Limitations**: While some limitations are mentioned, a more detailed discussion regarding the constraints of the methodology and potential biases in keyword selection is needed.

5. **Implications and Future Research**: Expand on the practical implications of the findings. Also, suggest specific areas for future research based on the trends observed.

6. **References and Citations**: Ensure all references are up-to-date and relevant. Some citations seem dated considering the fast-paced changes in the field.

#### Conclusion

The manuscript is promising but requires major revisions for clarity, depth, and broader impact. The suggested changes aim to enhance the overall quality and contribute significantly to the field of football research.

Reviewer #2: Dear Dr Jongwon Kim,

Thank you for the opportunity to review this paper. I have read the paper and made some comments based on my observations which are highlighted in this document.

It is clear you have conducted a large amount of research using some advanced techniques to identify the appropriate papers to be used within this study and the commitment to do so should be congratulated. However, from a scientific perspective, the methodology and statistical analysis techniques have not been explained thoroughly enough and with sufficient detail within the manuscript thus, the experimentation would be difficult to replicate.

For example, while formulas are provided for degree centrality and Eigenvector centrality, the components of these formulas are not fully explained. Each formula uses the ‘total number of nodes’ as a measure but an explanation of what the nodes are specifically is not included within the text which is confusing for the reader as it is the first time that nodes are mentioned in the manuscript. Furthermore, within the clustering analysis section, the terms ‘modularity coefficient’ and the ‘greedy heuristics algorithm of the hard clustering method’ are both mentioned without explanation or reference. In addition, explanations are not provided for other terms such as the Gini Coefficient, Ochiai coefficient and Salton coefficient either.

While the concept of using different analysis methods (network analysis) compared to more traditional review methods is an interesting one, I am unsure on the value the information from this manuscript provides to the scientific community. It is interesting to see how research trends have changed throughout the last 30 years, but this manuscript does not provide comprehensive insights into the reasoning behind these trends and how these may shape further research. The discussion needs further development to highlight the benefits of this study and to provide more in-depth insights on its value for the reader. The discussion needs to draw more on previous research to build these points.

I believe this manuscript has not been written with a proficient standard of the English language to be considered for publication in Plos One. There are frequent grammatical, punctual and tense based errors throughout the manuscript that make it difficult to read in places and in some instances, it does not fully make sense. These issues require attention before this manuscript can be considered ready for publication.

I hope the reasons highlighted in this document have provided some clarity as to how a decision was made regarding this manuscript. Please do not be disheartened by these comments as you have clearly developed a high-level skillset to conduct this research, but the manuscript needs further development before it is ready to be submitted for publication.

Kind regards,

Reviewer #3: Author should add the Temporal Analysis:

Analyzing the temporal evolution of football research trends allows us to trace the emergence and decline of specific topics over time. This can provide insights into the dynamic nature of football-related scholarship.

6. PLOS authors have the option to publish the peer review history of their article (what does this mean?). If published, this will include your full peer review and any attached files.

Reviewer #1: **Yes: **Mehrdad Agha Mohammad Ali Kermani

Reviewer #2: No

Reviewer #3: **Yes: **Prof. Dr. Muhammad Zafar Iqbal Butt

---

## [Author Response · Author response to Decision Letter 0]

23 Jan 2024

Thank you for the valuable feedback from the reviewers. I have incorporated the suggestions from the reviewers, and the paper has been improved accordingly. I have attached rebuttal letter for response to reviewers and Revised Manuscript with Track Changes. Please review the revised sections. Thank you very much.

---

## [Decision Letter · Decision Letter 1]

29 Jan 2024

PONE-D-23-39390R1Analysis of football research trends using text network analysisPLOS ONE

Dear Dr. KIM,

Thank you for submitting your manuscript to PLOS ONE. After careful consideration, we feel that it has merit but does not fully meet PLOS ONE’s publication criteria as it currently stands. Therefore, we invite you to submit a revised version of the manuscript that addresses the points raised during the review process.

We look forward to receiving your revised manuscript.

Kind regards,

Julio Alejandro Henriques Castro da Costa

Academic Editor

PLOS ONE

Journal Requirements:

Reviewers' comments:

Reviewer's Responses to Questions

**Comments to the Author**

1. If the authors have adequately addressed your comments raised in a previous round of review and you feel that this manuscript is now acceptable for publication, you may indicate that here to bypass the “Comments to the Author” section, enter your conflict of interest statement in the “Confidential to Editor” section, and submit your "Accept" recommendation.

Reviewer #1: All comments have been addressed

Reviewer #3: All comments have been addressed

2. Is the manuscript technically sound, and do the data support the conclusions?

Reviewer #1: Yes

Reviewer #3: Yes

3. Has the statistical analysis been performed appropriately and rigorously? 

Reviewer #1: Yes

Reviewer #3: Yes

4. Have the authors made all data underlying the findings in their manuscript fully available?

Reviewer #1: Yes

Reviewer #3: Yes

5. Is the manuscript presented in an intelligible fashion and written in standard English?

Reviewer #1: Yes

Reviewer #3: Yes

6. Review Comments to the Author

Reviewer #1: I have reviewed the revised version of "Analysis of football research trends using text network analysis" for Plos One. The authors have made commendable improvements based on the initial feedback. However, I suggest an essential enhancement: the manuscript should draw connections with key scientometric studies that have utilized Social Network Analysis (SNA).

To strengthen the manuscript's relevance and theoretical grounding, I recommend referencing the following studies:

1. https://doi.org/10.1016/j.eswa.2022.117853

2. https://doi.org/10.1111/sms.14019

3. https://doi.org/10.1016/j.joi.2011.05.007

These references are pivotal in aligning the study with foundational work in the field and clarifying its contribution to scientometric research.

Thank you for the opportunity to review this manuscript. I believe these additions will significantly enhance its impact.

Sincerely,

Reviewer #3: proposed changes are well incorporated, So, it is recommended for acceptance and for further publishing process.

7. PLOS authors have the option to publish the peer review history of their article (what does this mean?). If published, this will include your full peer review and any attached files.

Reviewer #1: **Yes: **Mehrdad Agha Mohammad Ali Kermani

Reviewer #3: **Yes: **Prof. Dr. Muhammad Zafar Iqbal Butt

---

## [Author Response · Author response to Decision Letter 1]

8 Feb 2024

Thank you for all the valuable feedback from all reviewers. I have incorporated the suggestions from the reviewers, and the paper has been improved accordingly. I have attached 'respond to reviewers' file to explain what I revised. Thank you very much.

---

## [Decision Letter · Decision Letter 2]

16 Feb 2024

Analysis of football research trends using text network analysis

PONE-D-23-39390R2

Dear Dr. KIM,

We’re pleased to inform you that your manuscript has been judged scientifically suitable for publication and will be formally accepted for publication once it meets all outstanding technical requirements.

Kind regards,

Julio Alejandro Henriques Castro da Costa

Academic Editor

PLOS ONE

Additional Editor Comments (optional):

Reviewers' comments:

Reviewer's Responses to Questions

**Comments to the Author**

1. If the authors have adequately addressed your comments raised in a previous round of review and you feel that this manuscript is now acceptable for publication, you may indicate that here to bypass the “Comments to the Author” section, enter your conflict of interest statement in the “Confidential to Editor” section, and submit your "Accept" recommendation.

Reviewer #1: All comments have been addressed

2. Is the manuscript technically sound, and do the data support the conclusions?

Reviewer #1: Yes

3. Has the statistical analysis been performed appropriately and rigorously? 

Reviewer #1: Yes

4. Have the authors made all data underlying the findings in their manuscript fully available?

Reviewer #1: Yes

5. Is the manuscript presented in an intelligible fashion and written in standard English?

Reviewer #1: Yes

6. Review Comments to the Author

Reviewer #1: (No Response)

7. PLOS authors have the option to publish the peer review history of their article (what does this mean?). If published, this will include your full peer review and any attached files.

Reviewer #1: **Yes: **It is me.

---

## [Editor Report · Acceptance letter]

27 Mar 2024

PONE-D-23-39390R2 

PLOS ONE

Dear Dr. KIM, 

I'm pleased to inform you that your manuscript has been deemed suitable for publication in PLOS ONE. Congratulations! Your manuscript is now being handed over to our production team.

Kind regards, 

on behalf of

Dr. Julio Alejandro Henriques Castro da Costa 

Academic Editor

PLOS ONE